# A Brief Review on the Synthesis of the *N*-CF_3_ Motif in Heterocycles

**DOI:** 10.3390/molecules28073012

**Published:** 2023-03-28

**Authors:** Zizhen Lei, Wenxu Chang, Hong Guo, Jiyao Feng, Zhenhua Zhang

**Affiliations:** 1College of Science, China Agricultural University, Beijing 100193, China; 2College of Plant Protection, China Agricultural University, Beijing 100193, China

**Keywords:** *N*-trifluoromethyl group, heterocycles

## Abstract

The trifluoromethyl group is widely recognized for its significant role in the fields of medicinal chemistry and material science due to its unique electronic and steric properties that can alter various physiochemical properties of the parent molecule, such as lipophilicity, acidity, and hydrogen bonding capabilities. Compared to the well-established C-trifluoromethylation, *N*-trifluoromethylation has received lesser attention. Considering the extensive contribution of nitrogen to drug molecules, it is predicted that constructing *N*-trifluoromethyl (*N*-CF_3_) motifs will be of great significance in pharmaceutical and agrochemical industries. This review is mainly concerned with the synthesis of heterocycles containing this motif. In three-membered heterocycles containing the *N*-CF_3_ motif, the existing literature mostly demonstrated the synthetic strategy, as it does for four- and larger-membered heterocycles. Certain structures, such as oxaziridines, could serve as an oxidant or building blocks in organic synthesis. In five-membered heterocycles, it has been reported that *N*-CF_3_ azoles showed a higher lipophilicity and a latent increased metabolic stability and Caco-2-permeability compared with their N-CH_3_ counterparts, illustrating the potential of the *N*-CF_3_ motif. Various *N*-CF_3_ analogues of drugs or bioactive molecules, such as sildenafil analogue, have been obtained. In general, the *N*-CF_3_ motif is developing and has great potential in bioactive molecules or materials. Give the recent development in this motif, it is foreseeable that its synthesis methods and applications will become more and more extensive. In this paper, we present an overview of the synthesis of *N*-CF_3_ heterocycles, categorized on the basis of the number of rings (three-, four-, five-, six- and larger-membered heterocycles), and focus on the five-membered heterocycles containing the *N*-CF_3_ group.

## 1. Introduction

Organic fluorine chemistry, as a prominent research area, has garnered significant attention for several decades and has also become essential to the evolution of many different but interconnected research fields [1]. The introduction of the fluorine group, especially the trifluoromethyl group, into organic compounds has become known as one of the most efficient methods for modulating molecular properties, for example, lipophilicity and metabolic stability [2]. Due to its potential utility, many methods have been studied extensively [3,4]. However, in contrast to the well-developed C-trifluoromethylations, the *N*-trifluoromethyl (*N*-CF_3_) motif has rarely been investigated to date. Considering the widespread dissemination of nitrogen (especially nitrogen-containing heterocycles) in drug molecules [5,6,7], constructing the *N*-CF_3_ motif in molecules is of great significance in pharmaceutical and agrochemical industries. Drug analogues and potential agents containing the *N*-CF_3_ motif are partially shown in Figure 1 [8,9,10,11,12,13].

Despite the great potential of the *N*-CF_3_ motif, the synthesis of this moiety and its relative chemistry have been rarely explored. The limited use of *N*-CF_3_ compounds is primarily due to the absence of scalable methods for their preparation [14,15,16]. Recently, thanks to the new reagents and methods, this motif has increasingly been featured in the literature.

In terms of constructing the *N*-CF_3_ motif in heterocycles, there are two main approaches. The first involves utilizing starting materials containing the *N*-CF_3_ motif to generate heterocycles directly, while the second strategy entails introducing the CF_3_ group via trifluoromethylation or fluorination of nitrogen-containing species. In this paper we review the construction of *N*-CF_3_ heterocycles on the basis of the size of the cycles, involving three-, four-, five-, six- and larger-membered heterocycles. This article covers as much literature as possible, from the 1960s to early 2023. The structures mentioned in this paper are shown in Figure 2.

## 2. Three-Membered Heterocycles

In previous reports, the synthesis of three-membered *N*-CF_3_ heterocycles mainly relied on starting materials containing the *N*-CF_3_ motif. In 1964, Logothetis reported the aziridination of *N*-CF_3_ imines **1a** in the presence of diazomethane, which obtained aziridine **2a** in the yield of 64% (Figure 1a) [17]. Subsequently, Coe et al. investigated the substructure scope of imines [18]. Unfortunately, poor selectivity was exhibited when R was replaced with *^i^*C_3_F_7_ (perfluoroisopropyl) or CF=CHCF_3_. When R = *^i^*C_3_F_7_, a mixture of three components in the ratio of **2b**:**2b″**:**2b′** = 6:9:1 was obtained and when R = CF=CHCF_3_, a mixture of two products was obtained in the ratio of **2d**:**2d′** = 71:29 by F shift and further CH insertion into a C-F bond. Kaupp et al. synthesized some stable triaziridines **4** which could be purified by fractional distillation by the irradiation of azimines **3** at room temperature (Figure 1b) [19]. However, not much attention has been paid to *N*-CF_3_ triaziridine. 

Compared to the structures mentioned above, diaziridine attracted more attention, and its synthesis could be divided into two strategies: Mitsch et al., while studying the reductive defluorination-cyclization of organic fluoronitrogens, found that diaziridine **6** could be generated from 1,1-bis(difluoramino)perfluoro-2-azapropan **5** in the presence of ferrocene (Figure 1c) [20]. Later DesMarteau et al. obtained another diaziridine **6** by nucleophilic cyclization when studying perfluoroalkanamine ions (Figure 1d) [21]. In the presence of CsF, CF_2_=NX (when X = F) yielded perfluoroalkanamine ion (CF_3_NF^−^), which underwent further reaction with another CF_2_=NX to form perfluoro-1-methylformamidine **8**. On the basis of previous work, DesMarteau group achieved synthesis of other diaziridines (when X = Br, Cl) through perfIuoroalkanamine ions [22,23]. Meanwhile, other electrophilic species, such as *N*-CF_3_ imine **1a**, could also be attacked by CF_3_NF^−^, leading to 3,3-difluoro-1,2-bis(trifluoromethyl)diaziridine [24].

Furthermore, perfluorinated oxaziridine **9** has drawn wide attention and its structure, synthesis and applications have been studied. Petrov and Resnati have already summarized the synthesis and reactivity of perfluorinated oxaziridines, in which *N*-CF_3_ oxaziridines are included [25]. In previous reports, the most common methods of synthesis were oxidative cyclization. In 1976, DesMarteau et al. reported that oxaziridine was obtained by the oxidation of a *N*-CF_3_ imines **1a** by CF_3_OOH [26]. This reaction was performed in two steps: addition, and further cyclization mediated by NaF. Later, different metal fluorides were investigated by the authors of [27] and KHF_2_ was found to be the most suitable reagent for the yield of oxaziridine **7** (the yield was up to 92%). This method was difficult and the starting materials were potentially explosive. In order to make the reaction safer and more convenient, different oxidants, such as hydrogen peroxide [28] and chlorine gas in the presence of metal carbonate [29], etc., have been developed, but these methods were still difficult. Finally, using meta-chloroperbenzoic acid (mCPBA) as the oxidant was determined to be a safer and more attractive choice (Figure 2) [30,31].

To date, *N*-CF_3_ aziridine, triaziridine and diaziridine have still gained less attention and their properties and applications are less-developed. On the other hand, there have been studies into the structure, properties, and applications of oxaziridine [25] (shown in Figure 2), such as its reaction with nucleophiles [32,33] and its use as an oxidant [34] or as building blocks. These studies are included in Section 4.2 of this paper.

## 3. Four-Membered Heterocycles

Similarly, the synthesis of four-membered *N*-CF_3_ heterocycles was based on starting materials containing the *N*-CF_3_ motif. 

The predominant approach to oxazetidine was [2+2] cycloaddition reaction. Trifluoronitrosomethane (CF_3_NO) was the most common starting material. In the 1950s, Barr and Haszeldine reported that CF_3_NO reacted with tetrafluoroethylene **10a** to give two products: an oxazetidine **11a** and a copolymer (consisting of two monomers in a 1:1 ratio) [35,36]. The oxazetidine predominated in this reaction at a high temperature (ca. 100 °C) and the copolymer at room temperature (Figure 3a). Since then, several halogenated tetrafluoroethylenes (CF_2_=CXY) have been investigated (Figure 3b) [37,38]. It was found that the formation of an oxazetidine and the copolymer from CF_3_NO occurred most readily with the olefins CF_2_=CF_2_, CF_2_=CHF (adducts were a 99:1 mixture of 3,3,4-trifluoro-2-trifluoromethyl-1,2-oxazetidine and 3,4,4-trifluoro-2-trifluoromethyl-1,2-oxazetidine), CF_2_=CFCl, and CF_2_=CCl_2_, and less readily with perfluoro-olefins containing more than two carbon atoms, or with ethylenes containing two or more vinylic hydrogens [38]. 

Meanwhile, it was reported that an attack on the substituted allene by CF_3_NO led to another form of oxazetidine. Banks et al. found that tetrafluoroallene **12** reacted with CF_3_NO to yield a complex mixture of oxazetidine **13** and **14** [39]. By adjusting the reaction conditions, the highest yields of oxazetidine **13** and **14** can reach 43% and 42%, respectively. They also found that compound **14** could be obtained (82% yield) when heating oxazetidine **13** with CF_3_NO. Later, Haszeldine et al. synthesized a series of oxazedines **16** with limited regioselectivity and stereoselectivity through the reaction of N, N-bistrifluoromethylamino-substituted allenes **15** with CF_3_NO (Figure 3b) [40,41]. 

In addition to CF_3_NO, there were other reagents used in [2+2] cycloaddition. For example, in 1986, Sundermeyer and co-workers found that CF_3_N=S=O could react with ketene to generate thiazetidin **17** (Figure 4a) [42]. Burger et al. reported that the reaction of CF_3_N=C=O and boranamine led to diazaboretidin **18** (Figure 4b) [43].

## 4. Five-Membered Heterocycles

The literature on three- and four-membered *N*-CF_3_ heterocycles was more focused on their synthesis, with limited exploration of their properties and potential applications. However, in contrast, five- and six-membered heterocycles featuring the *N*-CF_3_ motif have recently been receiving more and more attention [8,44,45,46]. Their synthesis methods, as well as biological activities and derivatization, are gradually being studied.

To evaluate the suitability of the *N*-CF_3_ motif on amines and azoles in drug design, Schiesser et al. synthesized a series of *N*-CF_3_ amines and azoles (shown in Figure 3), and determined their stability in aqueous media and other properties [45]. For example, the stability of *N*-CF_3_ analogues of known bioactive compounds (sulfamethoxazole derivative **19a**, tetracaine derivative **20a**, inhibitors of hedgehog pathway **21a** [47,48], inhibitors of methionine aminopeptidase **22a** [49], inhibitors of interleukin-1 receptor associated kinase 4 (IRAK4) **23a** [50], and sildenafil analogue **24a**) were studied and are shown in Table 1.

Two anilines **19a** and **20a**, as well as piperazine **24a**, showed fast hydrolysis at all three pH values investigated, with half-lives of less than 1.5 days at 25 °C (Table 1). It should to be noted that for **20a**, the corresponding carbamoyl fluoride **20b** was the main product at pH 1.0, with a small amount of product where both the carbamoyl fluoride had been further hydrolyzed to the secondary amine and the ester bond had been cleaved. The latter compound was also the main product at pH 7.4 and pH 10.0. In contrast to the *N*-CF_3_ anilines and piperazine, for all the *N*-CF_3_ azoles they investigated no corresponding carbamoyl fluoride of free azole was detected in aqueous media at any of the three pH values studied. 

Moreover, they then compared the key in vitro properties in medicinal chemistry (log *D*, experimentally determined polar surface area (ePSA) [51,52], permeability in human epithelial colorectal adenocarcinoma cells (Caco-2), and metabolic stability for the *N*-CF_3_ compounds **19a**–**24a** and their *N*-CH_3_ counterparts (Table 2). 

In the compounds investigated, the exchange of a methyl for trifluoromethyl led to the expected higher lipophilicity as proven by an increased log *D*_7.4_ and chromlog *D*_7.4_ and a decreased ePSA. Log *D*_7.4_ increases by on average 1.1 log units and chromlog *D*_7.4_ by 1.6 log units. However, the extent of this change can vary significantly and was dependent on both the individual compound and type of log *D*_7.4_ analysis used. Changes in permeability and metabolic stability were less consistent.Stability to human liver microsomes (HLMs) can be significantly increased for the trifluoromethyl analogue as seen for **22a** (*p* = 0.004) or decreased as for **21a**. The decreased metabolic stability of the latter two compounds could be due to an increased lipophilicity, rendering the potential metabolic soft spots (benzylic methyl group in **21a**) more susceptible to metabolism.

According to Schiesser’s research [45], *N*-CF_3_ amines were prone to hydrolysis, whereas *N*-CF_3_ azoles have excellent aqueous stability. Compared to N-CH_3_ analogues, *N*-CF_3_ azoles showed a higher lipophilicity and a latent increase in metabolic stability and Caco-2-permeability, which illustrated the value and potentiality of *N*-CF_3_ diazole in medicinal chemistry.

In terms of synthesizing these five-membered *N*-CF_3_ structures, both types of constructing *N*-CF_3_ were included. In this section, the synthesis of five-membered heterocycles would be divided into the three parts: nucleophilic fluorination, cyclization based on *N*-CF_3_ starting materials, and electrophilic trifluoromethylation. 

### 4.1. Nucleophilic Fluorination

#### 4.1.1. Fluorine/Halogen Exchange

Fluorine/halogen exchange was one of the first reactions to obtain the trifluoromethyl group on the nitrogen atom (Figure 5) [16]. Yagupolskii et al. achieved fluorine/halogen exchange with *N*-trichloromethyl derivatives in the presence of HF or Me_4_NF [53]. This strategy was also capable of generating *N*-CF_3_ pyrazoles [54,55] and *N*-CF_3_ 1,2,4-triazoles [55]. However, highly toxic or environmentally unfriendly reagents (such as CF_2_Br_2_, a known ozone-depleting reagent [56]) would be used in this method for fluorine/halogen exchange or in the preparation of the precursors (such as **26**) for fluorine/halogen exchange, which limited its use.

#### 4.1.2. Oxidative Desulfurization and Fluorination

Compared to fluorine/halogen exchange, this method has been much more thoroughly studied. This method allowed people to replace C-S bonds with C-F bods under extremely mild conditions compared to the fluorination of formamides [57] or fluorination induced by SF_4_ [58].

Hiyama et al. reported conversion from methyl dithiocarbamates to trifluoromethylamines in the presence of readily available fluoride ions (Figure 6a) [59]. The reaction conditions were applicable to a wide range of disubstituted nitrogen, with substituents including phenyl, heteroaromatic or alkyl. Recently, Schindler et al. applied chlorodithiophenylformiate as an electrophile and successfully obtained phenyl aminodithioate **30** [60]. Then trifluoromethylamines **31** were generated after the desulfurization and fluorination. Additionally, Hagooly et al. used BrF_3_ for desulfurization and fluorination and obtained 2-(trifluoromethyl)isoindoline-1,3-dione and 1-(trifluoromethyl)azepan-2-one [61]. 

Furthermore, Schoenebeck et al. reported that amines and SCF_3_^-^ source (Me_4_N)SCF_3_ could generate the highly electrophilic thiocarbomoyl fluoride **32** followed by a reaction with AgF to yield trifluoromethyl amines **31** [62]. Furthermore, there were other approaches to the intermediate **32**. Lin and Xiao et al. [63] generated this intermediate from difluorocarbene and sulfur (S_8_), while Jiang and Yi et al. [64] reported a method using CF_3_SO_2_Na (Figure 7). These strategies have been used to synthesize interesting analogues of biologically active compounds (examples are shown in Section 5.1).

#### 4.1.3. Cyclization Induced by Fluoride Ion

Fluoro-olefins containing terminal double bonds have been shown to isomerize in the presence of fluoride ion (Figure 8) [65]. There is a considerable amount of literature on the transformation of perfluoro-2,5-diazahexa-2,4-diene (CF_2_=NCF_2_CF_2_N=CF_2,_ **36a**) and its precursor CCl_2_=NCCl_2_CCl_2_N=CCl_2_, **35**. In 1967, Ogden and Mitsch reported that isomerization of perfluoro-, -diazomethines with CsF could form a cyclic five-membered *N*-CF_3_ heterocycle as a minor product (**34**, 30% yield) [66]. 

Scholl et al. synthesized **36a** from **35** via two steps, and found that **36a** could cyclize to form two isomers (**37a** and **37b**) in the presence of fluoride ion [67]. Both isomers could form nitrogen anion **37c** in the presence of fluoride ion. **37c** could also react with **37b** to generate substituted *N*-CF_3_ imidazolidine **38**. Subsequently, the same group found another transformation of **35** and obtained **39** in two steps [68]. In the presence of fluoride ion, another nitrogen anion **40** could be generated, which subsequently reacted with another **39** to obtain imidazolidine **41**.

Later in 1984, Banks et al. [69] investigated in some detail the effect of temperature and metal fluoride on the systems used by Scholl. The product composition depends on the reactivity of the fluoride source and the reaction conditions, i.e., the product may be under kinetic or equilibrium control. In addition to the substances reported earlier by Scholl, they detected others. Later, Chambers et al. [70,71] reacted the nitrogen anion **37c** with different trapping agents including haloalkane, perfluoro azaarene, perfluoro cyclobutene, etc., and obtained diverse heterocycles **38**. 

Meanwhile, Pawelke et al. [72] investigated the transformation of **35** or its derivative in the presence of fluoride source. Cyclization of compound **35** led to the perfluorinated 1H-imidazole **42**, whose chlorine atom could be further substituted by OPh or NEt_2_. 

### 4.2. Cyclization Based on N-CF_3_ Starting Materials

#### 4.2.1. [3+2] Cycloaddition

Utilization of starting materials containing the *N*-CF_3_ motif is a commonly employed strategy for achieving the target heterocyclic compounds. As mentioned in Section 2, perfluorinated oxaziridine **9a** could serve as building blocks in organic chemistry. DesMarteau et al. reported that some cycloaddition of oxaziridine **9a** with electron-rich alkenes and ketones resulted in oxazolidines or dioxazolidines (Figure 9) [73,74]. However, perfluorinated oxaziridine **9a** also had certain limitations as a building block: attempts to achieve the cycloaddition of **9a** with CH_2_=CH_2_, CFCl=CFCl, perfluorocyclopentene, acrylonitrile, and acetylene have failed. Additionally, the reaction could not occur with fluorinated ketones such as hexafluoroacetone. 

In recent decades [3+2] cycloaddition between azide and alkyne has attracted significant attention, and copper-catalyzed azide-alkyne cycloaddition is one of the most widely used forms of this technique [75]. In 2017, Beier et al. reported the synthesis of azidoperfluoroalkanes **43** which could be synthesized from perfluoroalkyl trimethylsilane (TMSR_F_) with p-toluenesulfonyl azide (TsN_3_) in the presence of CsF, or synthesized from (perfluoroethyl)lithium (R_F_ = C_2_F_5_) with TsN_3_ [76]. These azidoperfluoroalkanes could undergo [3+2] cycloaddition with alkynes catalyzed by copper to access diverse N-perfluoroalkyl 1,2,3-triazoles **44** (Figure 10). Later in 2018, Beier et al. reported a mild and efficient and synthesis of highly functionalized 1-perfluoroalkyl-1H-1,2,3-triazoles **45** via in situ generated enamines in azide-ketone [3+2] cycloaddition (Figure 10) [77].

In the same year, the Beier group developed a highly efficient method for the synthesis of a broad range of previously unreported N-fluoroalkyl-substituted five-membered heterocycles with microwave heating-assisted rhodium-catalyzed transannulation of N-fluoroalkyl-substituted 1,2,3-triazoles **44** [78]. Subsequently, the same group expanded this approach to acetylene substrates and successfully generated N-fluoroalkyl pyrrole (Figure 11) [79]. The mechanism proposed by authors is shown in Figure 12.

Xu, Guan, and Wang et al. [80] reported an alternative and scalable cyclization strategy based on *N*-CF_3_-containing synthons for constructing diverse *N*-CF_3_ azoles, including *N*-CF_3_ tetrazoles, *N*-CF_3_ imidazoles, and *N*-CF_3_ 1,2,3-triazoles (Figure 13). This method involved using a hypervalent iodine reagent for trifluoromethylation in combination with a base to efficiently carry out the reaction. Furthermore, estrone analogue **54** could be generated in two steps in a total of 74% yield. Subsequently, the authors’ group developed the reaction of the *N*-CF_3_ nitrilium ions **51** with N-, O-, and S-nucleophiles, resulting in various *N*-CF_3_ amidines, imidates, and Thioimidates [81]. Very recently, they utilized hypervalent iodine reagent for the trifluoromethylation of 4-alkylamino-pyridine to generate *N*-CF_3_ pyridinium salt which could be further translated to 2-functionlized nicotinaldehydes [82].

#### 4.2.2. Other Cyclization

In addition to [3+2] cycloaddition, other cyclization pathways have been explored for the synthesis of *N*-CF_3_-containing five-membered heterocycles. For instance, Sundermeyer and co-workers reported access to the preparation of imidazolidinedione **55** through the reaction of trifluoromethyl isocyanate with trimethylsilyl cyanide, followed by hydrolysis [83]. In 1977, Rudiger Mews synthesized dioxazolidine **56** from CF_3_NO and bis(trifluoromethyl)diazomethane [84]. Lentz reported that trifluoromethyl isocyanide reacted with hexafluoroacetone to yield compound **57** [85]. Later, the same group reacted trifluoromethyl isocyanide with diphosphene, leading to the formation of azaphospholidine **58** [86]. In addition, Crousse et al. developed a direct approach to obtaining *N*-CF_3_ hydrazines from CF_3_SO_2_Na. Among the family of *N*-CF_3_ hydrazines, hydrazides **59** showed hydrolysis in the presence of HCl and reacted further with diketone, leading to *N*-CF_3_-1H-pyrazoles **60** in 44% yield totally (Figure 14d) [87].

In 2019, Schoenebeck et al. reported straight access to *N*-CF_3_ amides, carbamates, thiocarbamates or ureas via *N*-CF_3_ carbomoyl building blocks **61** [88]. After that, the same group developed the transformation of the building blocks and generated non-cyclic or heterocyclic *N*-CF_3_ compounds as shown in Figure 15 [89,90,91,92]. Additionally, antihistamine derivative oxatomide analogue **64a** could be generated in 62% in two steps by N-H functionalization of **64**.

Meanwhile, Huang and Xu et al. reported the design and synthesis of novel *N*-CF_3_ hydroxylamine reagents **66** as well as their applications in preparation of *N*-CF_3_ compounds [93]. Some oxazolidinones **67** and **67′** could be generated from trifluoromethylamination/cyclization of styrenes or vinyl ether (Figure 16a). For example, estrone analogue **67a**, could be generated in 71% yield. Furthermore, ynamide **68** could be generated from reagent **66**, which could further form heterocycles **69** via Pd-catalyzed cyclization (Figure 16d). Subsequently, the same group employed reagent **66′** and trimethylsilyl cyanide to convert 1,3-enynes to trifluoromethylaminated allenes under a photoredox/copper-catalyzed 1,4-difunctionalization, in which allenes **70** could further yield oxazolidinones **71** in the presence of N-bromosuccinimide (NBS) or N-iodosuccinimide (NIS) (Figure 16e) [94].

### 4.3. Electrophilic Trifluoromethylation

In 2006, Togni et al. developed two new trifluoromethylation reagents (**72** and **73**) based on hypervalent iodine [95,96]. Subsequently, in 2011, the same group reported a Ritter-type direct electrophilic trifluoromethylation at nitrogen atoms using hypervalent iodine reagent **72** and obtained *N*-CF_3_ benzotriazole **74** as a side product (Figure 17a) [97]. Subsequently, they further refined the method and successfully conducted the trifluoromethylation of a variety of heterocycles (Figure 17b) [98]. 

Meanwhile, Umemoto developed diverse derivatives of (trifluoromethyl)dibenzofuranylium that could generate CF_3_^+^ anion at a low temperature, which facilitated the electrophilic trifluoromethylation of primary, secondary, or aromatic amines (Figure 17c) [99]. For example, 1-(trifluoromethyl)indoline could be generated in 68% yield.

## 5. Six-Membered Heterocycles

From the existing literature, the synthetic methodologies of six-membered heterocycles were similar to the five-membered heterocycles containing the *N*-CF_3_ motif. Therefore, in this section, some examples are shown briefly.

### 5.1. Nucleophilic Trifluoromethylation

Similarly, oxidative desulfurization and subsequent fluorination was also an efficient way to achieve six-membered heterocycles containing the *N*-CF_3_ motif, such as piperazines and piperidines. For example, Tlili et al. [100] used carbon disulfide and (diethylamino)sulfur trifluoride (DAST) to generate thiocarbomoyl fluoride intermediate **32**, and then synthesized a series of *N*-CF_3_ piperazines. Borbas et al. employed DAST and NBS to generate *N*-CF_3_ morpholine while studying N-fluoroalkylated nucleoside analogues [101]. In addition to AgF, pyridinium poly(hydrogen fluoride [8] could also be employed for oxidative desulfurization and fluorination to give the product **31c** which exhibited antibacterial activity. These methods allowed the introduction of the CF_3_ group into the nitrogen of pharmaceuticals or their analogues, demonstrating the potential of bioactive molecule modification (Figure 18) [8,62,64].

### 5.2. [4+2] Cycloaddition

As described in Section 3, CF_3_NO could serve as building blocks in cycloaddition. The application of CF_3_NO in [4+2] cycloaddition has been investigated [39,102,103,104,105,106,107], yielding various adducts **75** (Figure 19a). Carson et al. have studied the nucleophilic displacements of fluorine atoms in perfluoro-1,2-oxazines, in particular amino-defluorination reactions [107]. It was established that perfluoro-(3,6-dihydro-2-methyl-2H-1,2-oxazine) reacted with ammonia at room temperature to give a mixture of 4- and 5-amino derivatives, while when reacted with disubstituted amines in diethyl ester at −78 °C it only gave 5-amino compounds. Furthermore, other starting materials featuring *N*-CF_3_ were utilized in [4+2] cycloaddition, as shown in Figure 19b,c [108,109]. 

### 5.3. Other Approaches

In 1976, Haszeldine et al. reported the formation of triazine via unsymmetrical carbodiimide intermediate, which was subsequently dimerized, trimerized or intramolecular cyclized (Figure 20a) [110]. Similarly, Mews et al. reported that R_F_N=S=O reacted with SO_3_, leading to the formation of sulfonimide [111]. The degree of oligomer depended on the size of the substituent, and at what time R_F_ = CF_3_, dimer and trimer were formed in the ratio of 3:1 (Figure 20b). 

Direct fluorination of hydrocarbons by fluorine gas was indeed also a method used to synthesize the corresponding fluorous compounds. Lagow et al. reported the synthesis of perfluoro highly-branched heterocyclic fluorine compounds by direct fluorination, and also reported that 1,4-bis(trifluoromethyl)piperazine **79** was highly generated in 85% yield (Figure 20c) [112].

## 6. Seven- and Larger-Membered Heterocycles

Perfluoro-2,5-diazahexane-2,5-dioxyl **80** could readily attack nitric oxide and hydrogen-atom donors, giving adducts, such as its monofunctional analogue, bis-trifluoromethyl nitroxide ((CF_3_)_2_NO) [113,114], which could readily react with fluoro-olefins. In some cases, the reaction of bis-trifluoromethyl nitroxide with a variable valence element compound led to an increase in the oxidation state of that element.

Banks et al. reported that an attack by dioxyl **80** on tetrafluoroethylene or hexafluoropropene led mainly to the formation of copolymers in, and also to a smaller number of adducts (**81a**, **81b**) [115]. It should be noted that the yield of **81b** could rise to 63% if the reactants were mixed at room temperature and at ca. 25 mmHg pressure. In addition, Banks et al. pointed out that the formation of adducts would require a gas-phase reaction [113]. Subsequently, Tipping et al. investigated the scope of the cycloadduct formation by using fluoroalkenes and a wide variety of hydrogen-containing alkenes [116]. Their report clearly illustrated the limitations of gas-phase reaction. Such reactions must be restricted to simple ethenes or halogenated propenes due to the possibility of hydrogen abstraction occurring.

Later, Tipping et al. synthesized mercurial **84** from dioxyl **80** and investigated the reaction of mercurial **84** with halogenated alkanes, acid chlorides, and dichlorosilanes [117]. The reaction of mercurial **84** with dichlorodimethylsilane resulted in the formation of silicon-containing heterocycle in 93% yield, while with l,l-dichlorosilacyclobutane, the spiro compound was isolated in 64% yield. On the other hand, Booth et al. reported that the dioxyl **80** could react readily by oxidative addition to [Pt(PPh_3_)_4_] or [IrCl(CO)L_2_] (L = PPh_3_, AsPh_3_, PMePh_2_) to afford the corresponding metal–nitroso complex containing a seven-membered chelate ring [118]. The resulting complexes were stable in air for several days or in N_2_ atmosphere for several months.

Meanwhile, Smith et al. reported the reaction of SO_2_ and SF_4_ with dioxyl **80** and obtained two heterocycles **82a** and **82b** (Figure 21) [119]. In neither case was a copolymer formed, something which differed from the results from the reaction of dioxyl **80** with tetrafluoroethene and hexafluoropropene in a previous report [113]. Compound **82a** slowly reacted with PPh_3_ at room temperature giving deoxidation products **83** and **83′**.

Moreover, in 2018, Beier et al. reported another strategy based on N-perfluoroalkyl 1,2,3-triazoles (Figure 22) [120]. A series of then-unknown N-perfluoroalkyl azepine derivatives were obtained via the aza-[4+3]-annulation of triazoles **44** with both (E)-1-subtituted and 2-substituted dienes. When silyloxy-substituted butadiene was employed, *N*-CF_3_ azepinone **86′** could be prepared.

## 7. Other Methods

In addition to the methods mentioned above, various other synthetic routes have been explored for the generation of heterocycles containing the *N*-CF_3_ motif. However, considering the involvement of multiple cyclic structures in these methods, their classification is challenging. Therefore, these approaches are described in this section.

In 1971, Ogden [121] reported a route to some perfluoroheterocyclic compounds via fluoride ion. In his work, tetrafluoroformaldazine **87** reacted with oxalyl fluoride and other carbonyl fluorides and obtained the heterocycles **88**, which could be further photolysis to smaller heterocycles **89** (Figure 23).

In early organic chemistry, pyrolysis was an effective tool used to study the composition and properties of substances. For example, Banks et al. found that pyrolysis of perfluoropiperidine or perfluoromorpholine led to the generation of *N*-CF_3_ pyrrolidine **90a** or *N*-CF_3_ oxazolidine **90b**, respectively [102,122,123,124]. The pyrolysis of perfluorooxazinane in platinum at 480 °C led to the formation of perfluoro-(1-methylazetidine) **91** in 73% yield and trace perfluoro-(1-methyl-2-pyrrolidone) **92**, while at 580 °C/19 mm, the yield of **91** and **92** changed to 48% and 24%, respectively (Figure 24a). 

Tatlow et al. investigated the fluorination of 4-methylpyridine in the presence of caesium tetrafluorocobaltate (CsCoF_4_) and obtained perfluoro-(1,3dimethylpyrrolidine) **93** and its analogue, together with a range of (per)fluoro-pyridine bearing CF_3_, CHF_2,_ and CH_2_F groups (Figure 24b) [125]. Similar to CsCoF_4_, CoF_3_ was a useful reagent for the preparation of a wide array of highly fluorinated organic molecules including open chain/cyclic aliphatics and aromatics. However, the high reactivity of CoF_3_ meant that most of these reactions were relatively unselective with poor functional compatibility [126]. 

In addition, electrochemical fluorination (ECF) was one of the most commonly used methods for the fluorination of nitrogen-containing materials [127]. In the past few decades, many different nitrogen-containing materials [128,129,130,131,132,133] have been used in the ECF (Figure 24c), but the yield was generally unsatisfactory (mostly < 20%) and side products were inevitable, which limited the scope.

## 8. Conclusions

Over the last decades, the chemistry of the *N*-CF_3_ motif has been weakly developed because of the limited approaches and an incompatibility with functionalized molecules. Very recently, some new simpler, safer, and powerful methods of obtaining this motif have been explored. In general, the existing literature mainly focuses on synthesis, with limited properties or applications. In three-membered heterocycles containing the *N*-CF_3_ motif, cycloaddition, reductive defluorination-cyclization, nucleophilic cyclization, and oxidative cyclization can reach the motif. Among them, properties and applications of oxaziridine were reported (oxidant or building blocks [25,34,73,74]), while other three-membered heterocycles were not reported. In four-membered heterocycles, cycloaddition was the predominant approach, while trifluoronitrosomethane was the most common starting material. Similarly, there were limited reports on the applications of four-membered heterocycles containing the *N*-CF_3_ motif. In five-membered heterocycles, Scheiesser et al. [45] studied, for example, the stability in aqueous media, lipophilicity and metabolic stability of various *N*-CF_3_ amines or azoles, illustrating the potential of the *N*-CF_3_ motif in medicinal chemistry. Generally, five-membered heterocycles containing this motif can be synthesized from nucleophilic fluorination, cyclization, and electrophilic trifluoromethylation. Furthermore, N-fluoroalkyl 1,2,3-triazoles could serve as the building blocks to access some other N-fluoroalkyl heterocycles [78,79,120]. In six-membered heterocycles, the synthetic approaches were similar. In larger-membered heterocycles, the perfluoro-2,5-diazahexane-2,5-dioxyl showed its potential in coordination chemistry. The dioxyl could react readily with [Pt(PPh_3_)_4_] or [IrCl(CO)L_2_} to form the corresponding metal–nitroso complex containing a seven-membered chelate ring which was stable in air for several days or in N_2_ atmosphere for several months [118]. 

Overall, the literature has concentrated on the synthesis of this motif in recent years, and research investigating its properties or applications is becoming more frequent. We believe that the chemistry of the motif will become more and more clear, thereby extending its fields of application. We expect that this review will help to inspire the development of new synthetic strategies or the application of certain structures.

## Data Availability

Not applicable.

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
