# Peer review of "A Brief Review on the Synthesis of the N-CF3 Motif in Heterocycles"

_molecules, 2023, doi:10.3390/molecules28073012_

Round 1
Reviewer 1 Report
A review by Z. Zhang et al. focuses on important compounds in the field of organofluorine chemistry containing N-perfluoroalkyl fragment. This literature review combines early works and a number of recent works. The selected material will be of interest to a wide range of chemists and may be recommended for publication in Molecules after minor revisions.
The most important wish is to add a description of recent studies in the field presented: Dalton Trans., 2011,40, 8569-8580 (https://doi.org/10.1039/C1DT10524H), Chem. Eur. J. 2021, 27, 10973 (https://doi.org/10.1002/chem.202101436), Org. Lett. 2022, 24, 12, 2393-2398 (https://doi.org/10.1021/acs.orglett.2c00647), Chem. Eur. J. 2023, 29, e202203248 (https://doi.org/10.1002/chem.202203248), Org. Chem. Front., 2022,9, 4549-4553 (https://doi.org/10.1039/D2QO00242F).
Before concluding the review, it would be appropriate to insert a brief section on the practical applications of the obtained N-perfluoroalkylated heterocycles. In particular, a description of the biological properties of the obtained substances, their use in coordination chemistry.
Author Response
Dear reviewer,
Many thanks for your kind handling and the critical reviewing of our manuscript.
Following reviews’ suggestion, and take your suggestion into account, modified the manuscript and with track changes and “Point-to-point response” are all submitted.
We will be grateful if this revision could be accepted by Molecules.
Yours sincerely,
Zhenhua

Reviewer 2 Report
In this manuscript, Zhang and co-workers reports the importance of N-CF3 group in heterocycles. The summary presents exclusively the presence of NCF3 moiety in various heterocyclic systems. The manuscript could be a good addition to literature however significant concerns are there that need attention.
1. Authors should mention the time scale they covered.
2. The abstract is too brief. authors should elaborate it while presenting their point of view on NCF3 moiety. They should also mention in the abstract that in what context they have focused on CF3 in terms of applications, e.g. synthetic chemistry, material science etc.
3. Title should be more elaborative as well as concise and illustrative of manuscript's contents.
4. Authors should be consistent in presenting the details. e.g. yield etc. In some cases they report while in some schemes these are missing.
5. Some unnecessary capitalization of letters is present. e.g. meta-Chloroperbenzoic etc.
6. Authors briefly presents the biological data for selected compounds. There should be description in the abstract etc and reasoning for the selection of few examples.
7. Authors should include some mechanistic details that explain the formation of various rings.
8. The abstract should also include the description of applications of different rings as authors included in the text.
9. My major concern is about the conclusive remarks. It is very brief and incomplete. Conclude your discussion in a more attractive way. Any review article is incomplete without the addition of future outlook, how authors see the field developing and contributing to the overall literature. Add more details here.
Author Response
Dear reviewer,
Many thanks for your kind handling and the critical reviewing of our manuscript.
Following reviews’ suggestion, and take your suggestion into account, modified the manuscript and with track changes and “Point-to-point response” are all submitted.
We will be grateful if this revision could be accepted by Molecules.
Yours sincerely,
Zhenhua Zhang

Round 2
Reviewer 2 Report
Authors have improved the manuscript. It could be accepted now.